# Unveiling Structural Memorization: Structural Membership Inference Attack for Text-to-Image Diffusion Models

## ABSTRACT

With the rapid advancements of large-scale text-to-image diffusion models, various practical applications have emerged, bringing significant convenience to society. However, model developers may misuse the unauthorized data to train diffusion models. These data are at risk of being memorized by the models, thus potentially violating citizens' privacy rights. Therefore, in order to judge whether a specific image is utilized as a member of a model's training set, Membership Inference Attack (MIA) is proposed to serve as a tool for privacy protection. Current MIA methods predominantly utilize pixel-wise comparisons as distinguishing clues, considering the pixel-level memorization characteristic of diffusion models. However, it is practically impossible for text-to-image models to memorize all the pixel-level information in massive training sets. Therefore, we move to the more advanced structure-level memorization. Observations on the diffusion process show that the structures of members are better preserved compared to those of nonmembers, indicating that diffusion models possess the capability to remember the structures of member images from training sets. Drawing on these insights, we propose a simple yet effective MIA method tailored for text-to-image diffusion models. Extensive experimental results validate the efficacy of our approach. Compared to current pixel-level baselines, our approach not only achieves state-of-the-art performance but also demonstrates remarkable robustness against various distortions.

## CCS CONCEPTS

• **Security and privacy** → **Privacy protections**; • **Computing methodologies** → **Computer vision**.

## KEYWORDS

Privacy protections, Membership Inference Attack, Text-to-image diffusion models

## 1 INTRODUCTION

In recent years, large models, especially diffusion models [14, 42, 44] have shown superior generative performance and found extensive application across various fields. Moreover, the advent of the text-to-image diffusion models [30] has facilitated the creation of high-quality, diverse text-conditional images. These models have

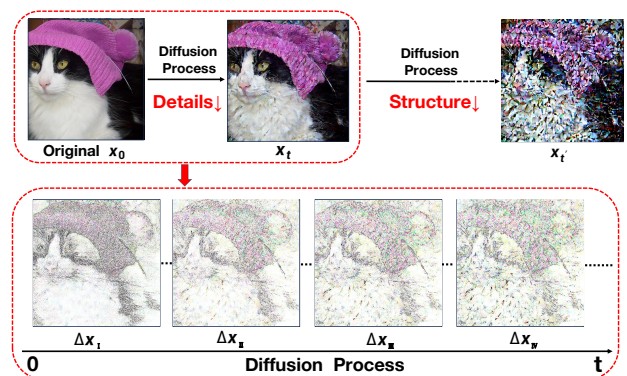

**Figure 1: Throughout the diffusion process, in the initial stage, diffusion models tend to corrupt the detailed features, whereas the overall image structure is preserved. The models continue to corrupt the image structure in the later stage.**

significantly propelled the advancements of Artificial Intelligence Generated Content (AIGC).

Nevertheless, the wide adoption of large models has raised various legal and ethical concerns, notably copyright issues [1], consent [8, 10] and ethics [17, 27]. One of the pressing concerns is the unauthorized use of images for training models. This not only risks compromising the privacy of image owners but also poses copyright infringements, as models can realistically replicate copyrighted artworks based on training data. This is attributed to models' capacity for memorization, which means models can remember certain elements or even reproduce almost identical images from their training datasets. Under such circumstances, Membership Inference Attack (MIA) [16] serves as an approach to tackle the issue. Given a specific unauthorized image, the goal of MIA is to determine whether it is a member of the training set of a target model. The core of MIA is to ingeniously exploit the models' memorization of members to distinguish them from non-members.

Recently, numerous Membership Inference Attack (MIA) methodologies [7, 19, 24] have been introduced for diffusion models. These methodologies, which rely on pixel-wise noise comparison, are designed to assess models' verbatim memorization of member images. However, we argue that it is practically impossible for large-scale text-to-image models to memorize all the pixel information, given that their training sets usually contain billions of images. For instance, the Stable Diffusion-v1-1 is trained on the LAION2B-en dataset, which contains around 2.32 billion text-image pairs. Hence, we attempt to capture more advanced memorization capabilities of large text-to-image diffusion models, specifically at the structure-level. To investigate the structure-level memorization, we first examine how a specific image is corrupted during the unidirectional diffusion process for better comprehension of image

structural variations, and then explore whether this correlates with models' memorization.

As illustrated in Figure 1, we iteratively employ noise to corrupt a specific image throughout the diffusion process. We then select various pairs of noisy images and compute the residuals between each pair. These residuals capture the change in image's corrupted parts. Our key observation is that diffusion models tend to corrupt the detailed features within the image in the initial diffusion stages, whereas the image structure is mostly preserved. Following this, the corruption extends to the overall structure of the image in the later diffusion stages. For instance, in Figure 1, the model primarily focuses on the detailed patterns of the hat in the very early stages. As the diffusion progresses, it then begins to address the structural aspects of the cat's fur. As for textual prompts in the text-to-image models, they primarily influence the overall structure and context of images in the later stages, while having minimal impacts in the early phases. Based on these findings, we delve deeper into the differences in structural corruption between members and nonmembers. We reveal that the structures of members are better preserved than those of nonmembers in the initial diffusion stages, as the diffusion models have memorized the structures of the members during training phases.

In light of the aforementioned observations, we introduce a straightforward yet effective MIA approach for text-to-image diffusion models by comparing the structural similarity between the original image and its corrupted version. Overall, the merits of our approach mainly contain three aspects: 1) Structural difference between members and nonmembers reveals the diffusion models' memorization at the skeletal level, which is preferred by large models. 2) Comparing differences at the image level is more robust to various distortions, particularly additional noise, than methods that rely on noise comparison. 3) Our method exhibits robustness to textual prompts, rendering it highly effective for membership inference tasks on images which lack training textual prompts in real-world scenarios.

We conduct a series of comprehensive experiments on both the Latent Diffusion Model and the Stable Diffusion under varying image resolutions. These experiments demonstrate the superior performance of our proposed method. Furthermore, we evaluate the robustness of our method under a range of practical distortions. Our findings confirm the resilience of our method. In addition, we examine the effect of diverse textual inputs on the efficacy of our method, as we can not obtain the ground-truth texts of images in training. Our results confirm that our method's performance is robust to changes in textual inputs, providing valuable insights to the practical application of MIA.

We summarize the contributions of this paper as follows:

- Instead of pixel-level memorization, we delve into the advanced memorization capabilities of large diffusion models at the structure-level. Furthermore, we investigate the differences in the preservation of image structures between members and nonmembers during the diffusion process.
- Drawing upon our findings, we propose a straightforward yet effective MIA method for text-to-image diffusion models by comparing the structural difference, which is more robust to various distortions.

- We further verify that our method exhibits robustness to variations in textual prompts, enabling its application to images lacking training textual prompts in real-world scenarios.
- Experimental results show that our method substantially outperforms existing MIA methods for text-to-image diffusion models, demonstrating its effectiveness.

## 2 RELATED WORK

### 2.1 Membership Inference Attack

As proposed by Shokri [37], Membership Inference Attack (MIA) aims to infer whether a specific sample is a member of a target model's training set. MIA is categorized into two main tasks: white-box attack and black-box attack. White-box attack [25, 29] presumes access to the internal structure and parameters of the target model, enabling a comprehensive analysis of the model's vulnerabilities. Conversely, black-box attack [32, 38, 41] operates solely through the model's observable inputs and outputs, posing a challenging yet more realistic scenario.

Primarily, MIA is specifically targeted at classification models [5, 23, 34, 45]. Subsequently, with the rapid development of generative models, an increasing number of MIA methods have begun to explore the vulnerabilities of such models, including VAE [18] and GAN [9]. For instance, LOGAN [11] is the first to adopt MIA to GAN in both white-box and black-box settings. It utilizes the outputs from the discriminator for inference in white-box scenario, while training a shadow GAN model in black-box scenario. Hilprecht et al. [12] proposes the Monte Carlo score and reconstruction loss, which can be used for attacking VAE. GAN-Leaks [3] also uses the Monte Carlo score for attacking GAN in black-box scenario.

**MIA for diffusion models.** Recently, several MIA methods targeting diffusion models have emerged. The Naive Loss method [24] and PIA [19] both use the training loss of diffusion models as a metric for membership inference, specifically by comparing the added noise with the predicted noise. The key difference is that Naive Loss method employs random Gaussian noise, whereas PIA utilizes the diffusion model's output at time t=0 as the noise. SecMI [7] compares the distance between two adjacent noisy images, which are generated through the diffusion process and the denoising process respectively. Nevertheless, these methods all rely on pixel-wise noise prediction, which are suboptimal in larger models and are vulnerable to real-world perturbations.

### 2.2 Diffusion Models

Starting from Denoising Diffusion Probabilistic Model (DDPM) [14, 39], generative diffusion models have gained significant attention in recent times and achieved remarkable breakthrough across diverse applications [6, 22, 28, 31, 33]. The training goal of diffusion models is to learn the reverse denoising process of gradually transforming Gaussian noise into signal. Score-based generative models train a neural network to forecast the score function, enabling the generation of samples through Langevin Dynamics [42–44]. The sampling process can either be a Markov process like DDPM, or a non-Markov process, such as DDIM [40]. Non-Markov process like DDIM can be used to accelerate the generating process.

Except for unconditional generation from pure noise, diffusion models have also been explored for conditional generation, such

as text-guided image generation. The text-to-image model [30] incorporates an image encoder-decoder framework to efficiently conduct the diffusion and denoising process within a latent space. The encoder compresses the input sample into a latent representation, while the decoder reconstructs the latent sample back to pixel space. Classifier guidance [6] and classifier-free guidance [13] are both proposed for high-quality image generation conditioned on various textual prompts.

## 2.3 Prior of Diffusion Generation Process

Although diffusion models have demonstrated superior generation performance, elucidating the generation process poses significant challenges. Until now, several researches have tried to explore and analyze the generation process. Choi et al [4] have conducted experiments on measuring the LPIPS distance of two different images under various time steps. They conclude that diffusion models learn coarse features and structures when the Signal-to-Noise Ratio (SNR) is low, whereas they learn more subtle and imperceptible features as the SNR becomes higher. Based on their observations, Wang et al. [46] design an encoder to provide comparatively strong conditions for the diffusion model when the SNR is below $5e^{-2}$ in the super-resolution image generation task. Likewise, Kwon et al. [20] also verify that modification of the generation process in the early denoising stage can achieve larger high-level semantic changes. Furthermore, Park et al. [26] conduct exponential sampling to carry out an analysis of the generation process. They conclude that in the early denoising stage, the diffusion models establish spatial information representing semantic structure, and then widen to the regional details of the elements in the later stage.

## 3 METHOD

Given an image $x_0$, our goal is to infer whether $x_0$ belongs to the training set of a diffusion model $\epsilon_\theta$. Current methods mainly leverage pixel-level memorization. We argue that for large-scale model, its memory mechanism is beyond pixel-level to structure-level. To demonstrate this, we first explore the structural changes throughout the diffusion process. We find that the structural information is largely maintained in the initial steps, and the members' structures are better preserved as the diffusion models have seen the structures of members during the training process (Section 3.2). Based on this observation, we design a structure-level MIA for text-to-image diffusion models (Section 3.3). The overview of our proposed method is shown in Figure 3.

## 3.1 Preliminaries

**Text-to-Image Diffusion Models.** Distinct from other traditional generative models, diffusion models contain two processes: the diffusion (forward) process and the denoising (backward) process. During the diffusion process, diffusion models iteratively introduce Gaussian noise to the original image $x_0$ with a total steps of T:

$$q(x_{1:T}|x_0) = \prod_{t=1}^{T} q(x_t|x_{t-1}) \tag{1}$$

where:

$$q(x_t|x_{t-1}) = \mathcal{N}(x_t; \sqrt{1-\beta_t}x_{t-1}, \beta_t \mathbf{I}) \tag{2}$$

and the variance schedule $\beta_1, ..., \beta_T$ is predefined. As t approaches T, $\beta_t$ becomes closer to 1.

During the denoising process, diffusion models generate image through multiple denoising steps starting from Gaussian noise:

$$p(x_{0:T}) = p(x_T) \prod_{t=1}^{T} p_\theta(x_{t-1}|x_t) \tag{3}$$

where:

$$p_\theta(x_{t-1}|x_t) = \mathcal{N}(x_{t-1}; \mu_\theta(x_t, t), \Sigma_\theta(x_t, t)) \tag{4}$$

and $\Sigma_\theta(x_t, t)$ is a constant depending on $\beta_t$, $\mu_\theta(x_t, t)$ is predicted by a neural network $\epsilon_\theta$ as:

$$\mu_\theta(x_t, t) = \frac{1}{\sqrt{\alpha_t}}\left(x_t - \frac{\beta_t}{\sqrt{1-\bar{\alpha}_t}}\epsilon_\theta(x_t, t)\right) \tag{5}$$

Under this formulation, in text-to-image diffusion models, we use classifier-free guidance [15] to guide the image generation by textual prompts y. The degree of text influence is controlled by adopting Eq.6 and adjusting the unconditional guidance scale $\gamma$.

$$\epsilon_\theta(x_t|y) = \epsilon_\theta(x_t|\emptyset) + \gamma \cdot (\epsilon_\theta(x_t|y) - \epsilon_\theta(x_t|\emptyset)) \tag{6}$$

**DDIM Inversion.** To expedite the denoising process and ensure a unique output, deterministic DDIM sampling [40] has been introduced, thereby enabling a skip-step strategy. Then for the diffusion process, a simple inversion technique, named DDIM inversion, has been suggested for the DDIM sampling. Such inversion process in Eq.7 provides a deterministic transformation between an input image and its corrupted version.

$$x_{t+1} = \sqrt{\alpha_{t+1}}\left(\frac{x_t - \sqrt{1-\alpha_t}\epsilon_\theta(x_t, t)}{\sqrt{\alpha_t}}\right) + \sqrt{1-\alpha_{t+1}}\epsilon_\theta(x_t, t) \tag{7}$$

We also give more mathematical details in Supplementary Materials.

## 3.2 Structure Evolution in Diffusion Process

To better capture the structure-level memorization of diffusion model, we first explore the changes in structure information throughout the diffusion process. Current arts [4, 20, 26, 46] show that during image generation, diffusion models focus more on imperceptible details when the noise levels are minimal, while concentrating on high-level context when faced with high noise levels. Similarly, but more carefully, we especially focus on the changes in structural information of both members and non-members throughout the unidirectional diffusion process. We leverage the structural similarity (SSIM) [47] as a metric. During the diffusion process, the original image $x_0$ is gradually corrupted by noise. A lower SSIM between $x_0$ and its corrupted version $x_t$ indicates greater structural loss. We first explore the **decrease rate ($v$)** of SSIM throughout the whole diffusion process for both members and nonmembers:

$$v(t) = \frac{SSIM(x_0, x_{t+\triangle t}) - SSIM(x_0, x_t)}{\triangle t} \tag{8}$$

Figure 2 (a) depicts the average decrease rate over 500 members and 500 nonmembers. It can be noted that the rate of decrease in SSIM between original images and its corrupted version is observed to initially increase and then decrease. More significantly, the decrease rates for members and nonmembers exhibit distinct behaviors. Nonmembers exhibit a higher rate of decrease when the diffusion timestep t ranges from 0 to approximately 100. This

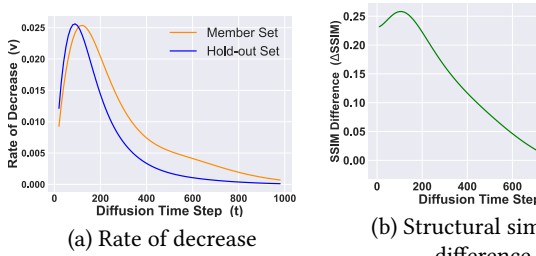

(a) Rate of decrease        (b) Structural similarity difference

Figure 2: (a) The decrease rate of structural similarity for the member set and the hold-out set. The structural similarity exhibits a steeper decline for images belonging to the hold-out set during the initial diffusion stage. (b) The average difference in the structural similarity between the member set and the hold-out set. The structural similarity for the member set surpasses that for the hold-out set during the first 800 diffusion steps, peaking at around step 100.

suggests that, for images that have been exposed to the diffusion models during training, their structures are more apt to be maintained in the early diffusion steps compared to images that are not included in the training set. However, as the images are further corrupted, the structural information is diluted by noise. The rate of decrease in structural similarity for members is even greater than that for non-members.

Given the difference in decrease rate among members and non-members, we further assess the average **SSIM difference ($\triangle SSIM$)** between the member set $D_m$ and the hold-out set $D_h$:

$$\triangle SSIM(t) = \frac{1}{|X_m|} \sum_{x_0 \in X_m} SSIM(x_0, x_t) - \frac{1}{|X_h|} \sum_{x_0 \in X_h} SSIM(x_0, x_t) \tag{9}$$

where $X_m \sim D_m$, $X_h \sim D_h$. Figure 2 (b) depicts the average SSIM difference over 500 members and 500 nonmembers. The structural similarity for the member set is larger than that for the hold-out set in the first 800 diffusion steps. Besides, the difference in structural similarity between the member set and the hold-out set gradually increases during the first 100 diffusion steps, reaching a maximum at around step 100, which serves as an important clue for dividing member set images and hold-out set images. These findings offer a foundation for our proposed straightforward MIA strategy.

## 3.3 Structure-Based Membership Inference Attack

Following the intuition above, we introduce a simple yet effective membership inference attack method for text-to-diffusion models, centered on the structure similarity between the original image and its corrupted version.

As shown in Figure 3, we input an image $x_0$ into the encoder of the text-to-image diffusion model, thereby obtaining its latent representation $z_0$. We also adopt the BLIP [21] model to extract a caption from image as textual prompt, since in practical applications, it is difficult to obtain the training-time texts corresponding to the images. Then we follow Eq. 7 to perform DDIM inversion to $z_0$ in

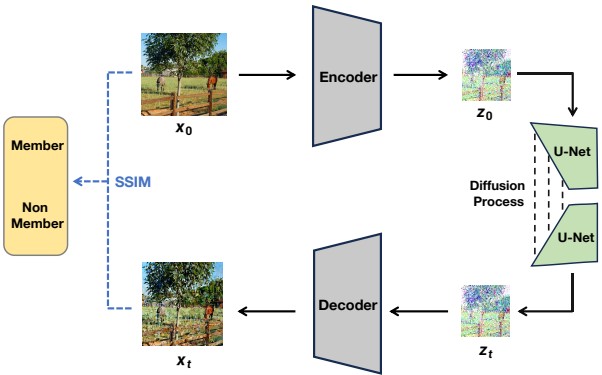

Figure 3: An overview of our proposed method. Given an input image, we first utilize the encoder of the text-to-image diffusion model to transform it to its latent representation $z_0$. Then we conduct DDIM inversion in the diffusion process, and get the noisy latent $z_t$. Next, we leverage the decoder of the diffusion model to transform $z_t$ back to the pixel space, thereby obtaining the output image. Finally, we compare the structural similarity between the input and the output to determine whether the input image belongs to the training set of the diffusion model.

the latent space, and get the corrupted latent $z_t$. Subsequently, we utilize the decoder of the text-to-image diffusion model to transform $z_t$ back to the pixel space and get $x_t$. The ingenious application of the encoder and decoder in the text-to-image model enables image-level comparison, facilitating the extraction of intricate structures without noise interference. By computing the structural similarity (SSIM) between $x_0$ and $x_t$, we obtain a membership score for $x_0$ and predict its membership as the following:

$$x_0 = \begin{cases} \text{member,} & \text{if } SSIM(x_0, x_t) > \tau \\ \text{nonmember,} & \text{if } SSIM(x_0, x_t) \leq \tau \end{cases} \tag{10}$$

This indicates that we consider an image is a member of the training set of the target model $\theta$ if $SSIM(x_0, x_t)$ is larger than a threshold $\tau$.

## 4 EXPERIMENTS

### 4.1 Experimental Setup

**Target Models and Datasets.** We utilize two prominent text-to-image diffusion models: the Latent Diffusion Model and the Stable Diffusion-v1-1, trained on the LAION-400M [36] and LAION2B-en [35] datasets, respectively. We conduct experiments on the two models without further fine-tuning or other modifications. For the datasets, the LAION-400M dataset comprises 400 million text-image pairs, while LAION2B-en, a subset of LAION-5B, contains approximately 2.32 billion English text-image pairs. These datasets are crawled from the Internet which are general and diversified. Additionally, we employ the COCO2017-Val dataset, which includes 5,000 images and is commonly adopted for model evaluation.

**Table 1: Performance of our proposed method and baseline methods on the Latent Diffusion Model, with resolutions 512 and 256. ↑ represents that the higher the metric, the better the performance. Bold denotes the best result for each metric.**

| | 512×512 | | | | 256×256 | | | |
|---|---|---|---|---|---|---|---|---|
| Method | AUC↑ | ASR↑ | Precision↑ | Recall↑ | AUC↑ | ASR↑ | Precision↑ | Recall↑ |
| SecMI | 0.759 | 0.699 | 0.749 | 0.620 | 0.732 | 0.680 | 0.754 | 0.557 |
| PIA | 0.656 | 0.655 | 0.789 | 0.420 | 0.725 | 0.695 | **0.822** | 0.505 |
| NaivelLoss | 0.789 | 0.740 | 0.830 | 0.605 | 0.737 | 0.709 | 0.815 | 0.553 |
| Ours | **0.930** | **0.860** | **0.880** | **0.839** | **0.841** | **0.763** | 0.799 | **0.720** |

**Table 2: Performance of our proposed method and baseline methods on the Stable Diffusion, with resolutions 512 and 256. ↑ represents that the higher the metric, the better the performance. Bold denotes the best result for each metric.**

| | 512×512 | | | | 256×256 | | | |
|---|---|---|---|---|---|---|---|---|
| Method | AUC↑ | ASR↑ | Precision↑ | Recall↑ | AUC↑ | ASR↑ | Precision↑ | Recall↑ |
| SecMI | 0.712 | 0.671 | 0.725 | 0.552 | 0.681 | 0.643 | 0.728 | 0.479 |
| PIA | 0.623 | 0.636 | 0.816 | 0.357 | 0.725 | 0.678 | **0.814** | 0.464 |
| NaivelLoss | 0.766 | 0.717 | 0.816 | 0.571 | 0.738 | 0.693 | 0.799 | 0.523 |
| Ours | **0.920** | **0.852** | **0.872** | **0.826** | **0.811** | **0.750** | 0.800 | **0.670** |

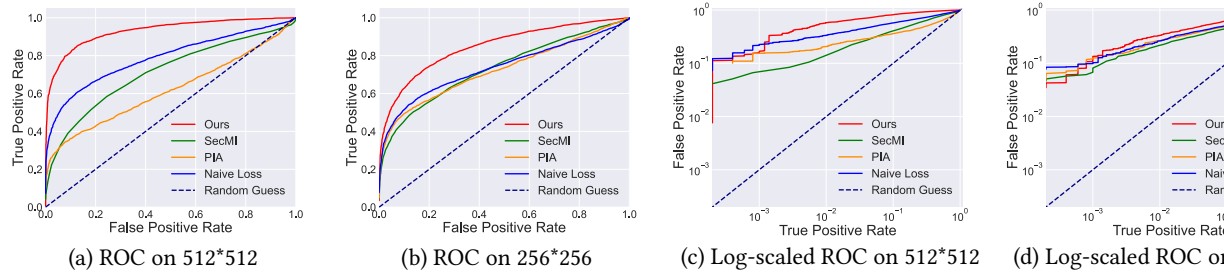

(a) ROC on 512*512    (b) ROC on 256*256    (c) Log-scaled ROC on 512*512    (d) Log-scaled ROC on 256*256

**Figure 4: The ROC and log-scaled ROC curves on the Latent Diffusion Model, with resolutions 512 and 256. The ROC and log-scaled ROC indicate that our method is significantly more effective on the Latent Diffusion Model compared to baselines.**

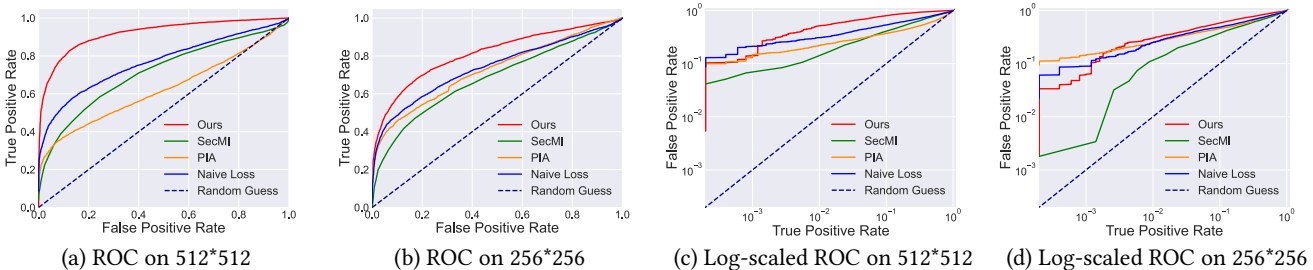

(a) ROC on 512*512    (b) ROC on 256*256    (c) Log-scaled ROC on 512*512    (d) Log-scaled ROC on 256*256

**Figure 5: The ROC and log-scaled ROC curves on the Stable Diffusion, with resolutions 512 and 256. The ROC and log-scaled ROC indicate that our method is significantly more effective on the Stable Diffusion Model compared to baselines.**

**Implementation Details.** For the two target models, we both use the 5000 images in COCO2017-Val as the hold-out set. As for member set selection, we randomly sample 5000 images from the LAION-400M dataset as the member set for the Latent Diffusion Model; and we also randomly sample 5000 images from the LAION2B-en dataset as the member set for the Stable-Diffusion-v1-1. Our experiments are conducted across two image resolutions: 256x256 pixels

and 512x512 pixels. Besides, we adopt DDIM inversion (Eq. 7) with an interval of 50 and incorporate noise addition twice during the forward diffusion process.

**Evaluation Metrics.** In order to evaluate the performance of our proposed method, we adopt the widely used metrics [16]: Attack Success Rate (ASR), Area-Under-the-ROC-curve (AUC), Precision and Recall. We also follow the metrics used in [2], including the

True Positive Rate (TPR) when the False Positive Rate (FPR) is 1% (TPR@1%), and the True Positive Rate when the False Positive Rate is 0.1% (TPR@0.1%). (More details about experimental setups can be found in Supplementary Materials)

**Table 3: The TPR at low FPR of our method and baselines on the Latent Diffusion Model, with resolutions 512 and 256.**

|  | 512×512 | | 256×256 | |
|---|---|---|---|---|
| Method | TPR@1%↑ | TPR@0.1%↑ | TPR@1%↑ | TPR@0.1%↑ |
| SecMI | 0.227 | 0.134 | 0.215 | 0.081 |
| PIA | 0.243 | 0.126 | 0.267 | 0.126 |
| NaivelLoss | 0.338 | 0.231 | 0.263 | 0.103 |
| Ours | **0.575** | **0.245** | **0.368** | **0.173** |

**Table 4: The TPR at low FPR of our method and baselines on the Stable Diffusion, with resolutions 512 and 256.**

|  | 512×512 | | 256×256 | |
|---|---|---|---|---|
| Method | TPR@1%↑ | TPR@0.1%↑ | TPR@1%↑ | TPR@0.1%↑ |
| SecMI | 0.138 | 0.067 | 0.108 | 0.003 |
| PIA | 0.219 | 0.152 | 0.247 | **0.147** |
| NaivelLoss | 0.310 | 0.215 | 0.250 | 0.114 |
| Ours | **0.512** | **0.234** | **0.302** | 0.107 |

## 4.2 Comparison to Baselines

We compare our method with three current MIA methods for diffusion models, including PIA [19], SecMI [7], and Naive Loss [24]. We leave the details of baselines in the Supplementary Materials.
**Evaluation on Latent Diffusion Model.** Table 1 shows the results on the Latent Diffusion Model. Compared to baselines, our method exhibits remarkable performance enhancements, particularly for images with resolution 512, where it surpasses all baselines in AUC, ASR, Precision, and Recall metrics. Notably, it achieves a 14.1% increase in AUC and a 12.2% increase in ASR compared to the next best method. For images with resolution 256, our method still outperforms baselines in AUC, ASR, and Recall, albeit with a marginal decrease in Precision. The ROC curve and log-scaled ROC curve is depicted in Figure 4. We also consider the TPR at very low FPR, i.e. 1% and 0.1% FPR, as shown in Table 3. Our method consistently outperforms in all assessments, underscoring its superiority in MIA performance. Particularly, its effectiveness significantly increases for images with resolution 512. This reveals large-scale model's structure-level memorization and highlights the potential of our method for more precise MIA on high-resolution images.
**Evaluation on Stable Diffusion.** Table 2 shows the results on the Stable-Diffusion-v1-1. Our method significantly exceeds baselines in AUC, ASR, and Recall. For images with resolution 512, it shows a 15.4% improvement in AUC and a 13.2% improvement in ASR over the nearest competitor. For images with resolution 256, our approach maintains a 7.3% higher AUC and a 5.5% higher ASR than the second-best method, despite a slight 1.4% reduction in Precision. Besides, the ROC curve and log-scaled ROC curve is depicted in Figure 5. The TPR at 1% FPR and 0.1% FPR is illustrated in Table 4. The results consistently demonstrate our method's ability to

**Table 5: The performance of our proposed method with different total timesteps on the Latent Diffusion Model, with resolution 512.**

| Total Timestep | AUC↑ | ASR↑ | Precision↑ | Recall↑ |
|---|---|---|---|---|
| 50 | 0.923 | 0.854 | 0.867 | 0.837 |
| 100 | 0.930 | 0.860 | 0.88 | 0.839 |
| 200 | 0.929 | 0.861 | 0.913 | 0.803 |
| 300 | 0.900 | 0.836 | 0.884 | 0.776 |
| 400 | 0.850 | 0.781 | 0.854 | 0.684 |
| 600 | 0.723 | 0.676 | 0.809 | 0.466 |
| 800 | 0.487 | 0.503 | 0.589 | 0.083 |

**Table 6: The performance of our proposed method with various sampling intervals on the Latent Diffusion Model, with resolution 512.**

| Interval | AUC↑ | ASR↑ | Precision↑ | Recall↑ |
|---|---|---|---|---|
| 1 | 0.933 | 0.863 | 0.911 | 0.806 |
| 10 | 0.934 | 0.864 | 0.885 | 0.838 |
| 20 | 0.933 | 0.864 | 0.897 | 0.824 |
| 50 | 0.930 | 0.860 | 0.880 | 0.839 |
| 100 | 0.924 | 0.855 | 0.881 | 0.835 |

produce high-confidence predictions across the Stable Diffusion by leveraging large-scale model's structure-level memorization.

## 4.3 Analysis of Total Timestep and Interval

**Total Timestep.** To evaluate the impact of the hyper-parameter total diffusion timestep T, we vary T from 50 to 800, with a fixed interval ($t_i$=50). For instance, setting T=200 involves adding noise from t=0 to t=200 in 50-step increments, totaling 4 query times. Experiments are conducted using the Latent Diffusion Model on images with resolutions 512 and 256. Results are shown in Table 5. AUC and ASR metrics remain stable between T=50 and T=200, then begin to decrease from T=300, continuing to drop with further increases in T. Notably, at T=800, AUC falls below 50%. The outcomes align with our findings outlined in Section 3.2. With total diffusion timesteps under 300, the model maintains the structural integrity of member images more effectively, distinguishing them from non-member images. As T increases over 300, noise accumulation adversely affects the structures of both member and non-member images, thus reducing the attack effectiveness.
**Interval.** To investigate the influence of the hyper-parameter interval $t_i$, we fixed the total timestep at 100 and varied $t_i$ from 1 to 100. Using the Latent Diffusion Model, we conducted experiments on images with resolutions 512 and 256. As demonstrated in Table 6, there is minimal variation in AUC and ASR across different $t_i$ settings, possibly due to our method's reliance on a deterministic diffusion process that eliminates random noise in each step. Thus, changes in $t_i$ do not significantly affect the image's structural information. Nonetheless, a higher $t_i$ value within the fixed total timestep implies more queries and higher computational costs. Therefore, we opt for $t_i$=50 as a practical compromise.

**Table 7: Performance of our method and baselines under distortions on the Latent Diffusion Model, with resolution 512.**

| Method | Noise | | Rotation | | Saturation | | Brightness | |
|---|---|---|---|---|---|---|---|---|
| | AUC↑ | ASR↑ | AUC↑ | ASR↑ | AUC↑ | ASR↑ | AUC↑ | ASR↑ |
| SecMI | 0.566 | 0.565 | 0.607 | 0.596 | 0.703 | 0.670 | 0.709 | 0.667 |
| PIA | 0.399 | 0.517 | 0.542 | 0.600 | 0.621 | 0.644 | 0.629 | 0.620 |
| NaivelLoss | 0.517 | 0.559 | 0.710 | 0.678 | 0.776 | 0.722 | 0.810 | 0.747 |
| Ours | **0.710** | **0.694** | **0.899** | **0.823** | **0.883** | **0.815** | **0.840** | **0.774** |

**Table 8: The TPR at low FPR of our method and baselines under distortions on the Latent Diffusion Model, with resolution 512.**

| Method | Noise | | Rotation | | Saturation | | Brightness | |
|---|---|---|---|---|---|---|---|---|
| | TPR@1%↑ | TPR@0.1%↑ | TPR@1%↑ | TPR@0.1%↑ | TPR@1%↑ | TPR@0.1%↑ | TPR@1%↑ | TPR@0.1%↑ |
| SecMI | 0.067 | 0.020 | 0.135 | 0.070 | 0.185 | 0.098 | **0.185** | 0.026 |
| PIA | 0.036 | 0.015 | 0.156 | 0.077 | 0.212 | 0.134 | 0.151 | 0.022 |
| NaivelLoss | 0.056 | 0.022 | 0.206 | 0.072 | 0.308 | **0.193** | 0.180 | 0.015 |
| Ours | **0.205** | **0.025** | **0.420** | **0.121** | **0.443** | 0.123 | 0.182 | **0.049** |

## 4.4 Robustness Evaluation

In real-world scenarios, images undergo various distortions, like noise and brightness fluctuations, during transmission. Additionally, augmentation techniques are often applied to modify training data for large-scale diffusion models, leading to the discrepancies between training images and their originals. This necessitates the robustness of our method to such variations.

We evaluate our method's robustness using the Latent Diffusion Model on images with resolution 512. Four degradation techniques are applied to images:

- **Additional noise.** Salt and Pepper Noise, which randomly corrupts 10% of the pixels in each image, is added to images.
- **Rotation.** Images are rotated by 10 degrees counterclockwise around the geometric midpoint.
- **Saturation.** The saturation levels of images are adjusted, either increased or decreased by 50%, with equal probability.
- **Brightness.** The saturation levels of images are altered, either increased or decreased by 50%, with equal probability.

Results are shown in Table 7 and Table 8. It is evident that our methods achieve the highest results in ASR, AUC and TPR at 1% FPR across all four types of distortions. Notably, our structure-level approach exhibits exceptional resilience against additional noise, whereas other baseline methods experience a significant decline in performance. This is attributed to their reliance on noise-level comparison, which renders them vulnerable to such disturbances. Collectively, these experimental results underscore the superior stability and robust nature of our methods in effectively handling diverse distortions.

## 4.5 Comparison to Backward Reconstruction

All the baseline MIA methods for diffusion models involve both the forward diffusion process for noise introduction, and the backward denoising process for noise prediction. On the contrary, our method only leverages the forward diffusion process. We argue that during the initial diffusion process, as the structures of nonmember images are more severely corrupted than those of members, the structural

**Table 9: The performance of backward reconstruction and forward diffusion (ours) on the Latent Diffusion Model, with resolutions 512 and 256.**

| 512×512 | | | | |
|---|---|---|---|---|
| Method | AUC↑ | ASR↑ | TPR@1%↑ | TPR@0.1%↑ |
| Backward Reconstruction | 0.907 | 0.834 | 0.398 | 0.147 |
| Forward Diffusion | **0.930** | **0.863** | **0.575** | **0.245** |
| 256×256 | | | | |
| Method | AUC↑ | ASR↑ | TPR@1%↑ | TPR@0.1%↑ |
| Backward Reconstruction | 0.824 | 0.753 | 0.276 | 0.109 |
| Forward Diffusion | **0.841** | **0.769** | **0.368** | **0.173** |

differences between members and nonmembers have widened significantly. Conversely, the denoising process, which acts as the inverse of the diffusion process, reconstructs both the corrupted member images and nonmember images to their original states, which leads to the reduction in the image structural differences between members and nonmembers.

To validate this findings, we utilize the comparison of the original images and their backward reconstructed states for MIA, and make a contrast with our method. We conducted experiments using the Latent Diffusion Model on images with resolutions 512 and 256. The results are illustrated in Table 9. We observe that employing backward reconstruction results in a decrease in AUC and ASR by roughly 3%, regardless of the image resolutions. The TPR at 1% and 0.1% FPR also decrease to a large extent when using backward reconstruction. This reveals the superiority of our method in utilizing the unidirectional diffusion process for MIA, compared to other bidirectional methods.

## 4.6 The Impact of Texts on Structural Similarity

To evaluate the influence of texts on our method' performance, we delve into the impact of the unconditional guidance scale $\gamma$ using the Latent Diffusion Model on images with resolutions 512 and 256. We use classifier-free guidance to guide the image generation by textual prompts. The degree of textual influence is controlled by adopting Eq. 6 and adjusting $\gamma$. Specifically, setting $\gamma$ to 0 renders

**Table 10: The performance of our method with different unconditional guidance scale $\gamma$ on the Latent Diffusion Model, with resolution 512.**

| Scale $\gamma$ | AUC↑ | ASR↑ | Precision↑ | Recall↑ |
|---|---|---|---|---|
| 0.0 | 0.930 | 0.862 | 0.878 | 0.842 |
| 1.0 | 0.930 | 0.860 | 0.880 | 0.839 |
| 2.0 | 0.930 | 0.860 | 0.870 | 0.853 |
| 3.0 | 0.930 | 0.860 | 0.871 | 0.851 |
| 4.0 | 0.930 | 0.859 | 0.871 | 0.850 |
| 5.0 | 0.930 | 0.862 | 0.871 | 0.850 |

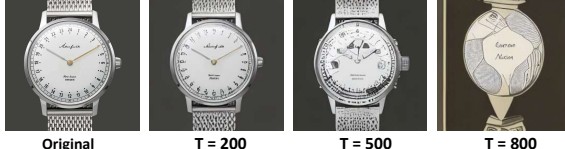

(a) BLIP: "A watch with a white dial and a silver mesh strap"

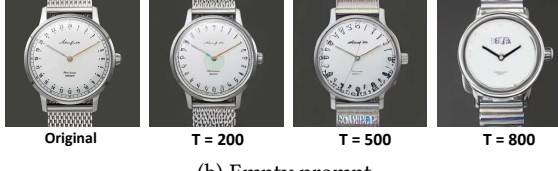

(b) Empty prompt

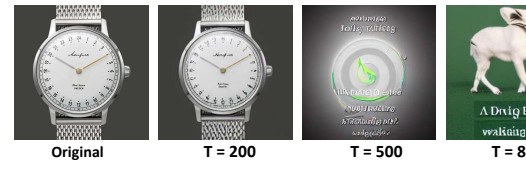

(c) Unrelated prompt: "A dog is walking on the grass"

**Figure 6: (a) Reconstruction results with a prompt extracted by BLIP model. (b) Reconstruction results with an empty prompt. (c) Reconstruction results with an unrelated prompt.**

the model entirely unconditional, while a setting of 1 makes it fully conditional, guided solely by text, forming the basis for our experiments. As $\gamma$ increases, the influence of textual information is increasingly pronounced. Here we vary $\gamma$ from 0 to 5. Results are shown in Table 10. All four metrics remain virtually unchanged across varying scale values, suggesting that textual information has minimal impact on structural similarity during the initial diffusion stage. The models preserve image structures well, irrespective of the presence of textual information when the noise levels are minimal.

To further explain this result, we conduct the reconstruction experiments, where we corrupt a image in the diffusion process to a certain timestep T, and then restore it in the denoising process. We compare the structural similarities between original images and their reconstructions under three conditions: captions from the BLIP model, empty prompts, and unrelated texts. One of the results is shown in Figure 6 (a) (b) (c). Notably, at T=200 (the noise level is low), reconstructions across all types of prompts are similar. However,

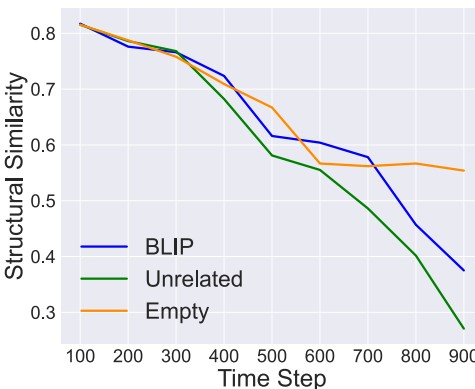

**Figure 7: The variation curve of structural similarity between the original image and the reconstruction image throughout total diffusion step T.**

as T increases to 500 and 800 (the noise levels are high), variations in reconstruction outcomes become pronounced. This indicates that textual impact on image structure is minimal at low noise levels, where models prioritize detailed features. Conversely, when faced with higher noise levels, where models focus more on overall structure, unrelated texts significantly influence the reconstruction results, underscoring the guiding role of textual information.

We also plot a trend curve depicting how structural similarity between the original and reconstructed images changes with the total diffusion step T. As shown in Figure 7, when T is below 300, structural similarity remains consistent across different prompts. However, as T increases, structural similarity experiences the sharpest declines with an unrelated prompt. These results suggest that the text impact on our method is minimal, since we only assess structural similarity in the initial diffusion phase where the noise levels are minimal.

## 5 CONCLUSION

In this paper, we explore the structure-level memorization of large-scale text-to-image diffusion models. We primarily investigate the corruption of images structures throughout the diffusion process. We further demonstrate that the structures of member images in training set are better preserved than those of nonmembers in the initial diffusion stages, since models can memorize member images' structures during training. Drawing on these insights, we introduce a novel Membership Inference Attack (MIA) method for text-to-image diffusion models to judge whether an unauthorized image is utilized for training a diffusion model. Our proposed method is to assess models' structure-level memorization. We evaluate our method on state-of-art text-to-image diffusion models, e.g., the Latent Diffusion Model and the Stable Diffusion. Experimental results show that our method achieves higher ASR, AUC, TPR @ 1% FPR and TPR @ 0.1% FPR than all baselines. Besides, our method also exhibits greater robustness against diverse distortions and maintains efficacy across different textual prompts, underscoring its applicability in more real-world contexts.

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
