# OpenReview forum: "Unveiling Structural Memorization: Structural Membership Inference Attack for Text-to-Image Diffusion Models"
_acmmm.org/ACMMM/2024/Conference — MM2024 Poster_

### Official Review · Reviewer_Jt1V · 2024-05-24

**Rating:** 3
**Confidence:** 3

**Summary:**

This paper addresses the infeasibility of pixel-level Membership Inference Attack (MIA) methods on large-scale models for determining whether a certain unauthorized image has been used in training. It proposes a novel MIA method for text-to-image diffusion model, which explores the structural-level memory of large diffusion models by studying the differences between member and non-member image structures during the diffusion process. By utilizing this structural-level memory, it achieves superior performance while also guaranteeing enhanced robustness.

**Strengths:**

(1)This method overcomes the limitations of traditional pixel-level MIA approaches, which rely on comparing pixel noise directly, when applied to large-scale models. By utilizing more advanced structural-level memory for MIA, it enables the attainment of excellent performance even within large-scale models.
(2)This method is more robust in dealing with various distortions of the image. Robustness experiments are conducted through four degradation techniques: adding additional noise, image rotation, modifying saturation, and modifying brightness. The experimental results show that the method advantages, especially when dealing with additional noise.

**Limitations:**

(1)The theoretical basis for the structural memory of diffusion models is insufficient when applied to MIA. The paper observes through experiments that during the diffusion process, the structural similarity of members will be greater than that of non-members. However, the paper does not provide a theoretical proof of the validity of this conclusion when applied to MIA. Simply relying on experimental observations may not be enough to fully prove the reliability and universality of this method in practical applications.
(2)The difference in structural similarity between members and non-members during the diffusion process lacks practical consideration. The difference in Structural Similarity Index (SSIM) peaks early in the diffusion process (around 0.25), and for Membership Inference Attacks (MIA) this difference needs to be large enough to reliably distinguish training set members from non-members. However, whether the difference of 0.25 is enough to achieve high-accuracy classification requires further analysis combined with a specific classification threshold τ.
(3)The paper overlooks an introduction to the stable diffusion model. While the experiments revolve around both the latent diffusion model and the stable diffusion model, the related work section only introduces concepts and principles pertaining to the latent diffusion model, completely omitting any discussion of the stable diffusion model.
(4)There are several issues present in the experimental setup of this paper. Firstly, the experimental methodology lacks clarity regarding the fundamental principles behind resolution selection. It seems inadequate to draw conclusions about the superiority of high-resolution images for MIA solely based on the general trend of resolution 512 outperforming resolution 256. Moreover, in real-world scenarios, resolution 512 may not be considered high resolution. Therefore, it remains uncertain whether higher or lower resolutions would impact structural memory and MIA effectiveness. Secondly, the experiment lacks a baseline for comparison. Simply comparing it to three methods does not sufficiently demonstrate its advantages. Thirdly, in the robustness experiment involving saturation modification, significant deviations from the optimal value are observed in the experimental results of this method when the false positive rate is 0.1%. What could be the underlying reason for this disparity?

**Suitability:**

2

---

### Official Review · Reviewer_i9na · 2024-05-24

**Rating:** 4
**Confidence:** 2

**Summary:**

This paper explores the advanced memorization capabilities of large diffusion models at the structure-level, investigating how these models preserve image structures differently between members (images used in the training set) and nonmembers during the diffusion process. Based on these findings, the paper proposes a Membership Inference Attack (MIA) method specifically tailored for text-to-image diffusion models, aiming to enhance privacy protection while maintaining robust performance.

**Strengths:**

S1.The paper is well organized and the conducts extensive experiments.

S2. This paper is overall well written. The problem is well formulated and clearly explained.

**Limitations:**

L1. The paper mentions the Latent Diffusion Model and the Stable Diffusion-v1-1, which are trained on the LAION-400M and LAION2Ben datasets, respectively, in Lines 454-456. Is it possible to conduct experiments with both models on both datasets?

L2. The structured MIA method may require significant computational resources to process the diffusion and denoising of images. It would be beneficial to include an analysis of the model's complexity to better understand its resource demands and efficiency.

L3. It would be great to open the code and dataset.

**Suitability:**

3

---

### Official Review · Reviewer_4i5a · 2024-05-25

**Rating:** 4
**Confidence:** 3

**Summary:**

The paper proposes a member inference attack for text-to-image diffusion models. Instead of pixel-level memorization, this paper studies the advanced memorization capabilities of large diffusion models at the structure-level, and the differences in the preservation of image
structures between members and non-members during the diffusion process.
Compared with existing advanced baselines, the proposed method is more robust to various image distortions and maintains efficacy across different textual prompts.

**Strengths:**

1、The paper proposes a novel perspective to study the member inference attacks from text-to-image diffusion models.
2、The ingenious application of the encoder and decoder in the text-to-image model enables
image-level comparison. The MIA attack method based on structural similarity is proposed,which is robust to image distortion.
3、Experimental results demonstrated the effectiveness of the proposed method.

**Limitations:**

1、The changes in structure information during the diffusion process have been studied by many
previous works [4, 20, 26, 46], thus this idea is not raised for the first time.
2、There are a large number of details that are not clearly explained. For example, there is no detailed description on how to set the threshold in Eq. (10).
3、The paper applies BLIP to obtain textual prompts, but it is not shown in Figure 3. Furthermore, there is no detailed explanation on how to utilize the textual prompts obtained through BLIP.
4、This paper does not provide source code to verify the feasibility of the method used in the paper.

**Suitability:**

3

---

### Official Review · Reviewer_p9fb · 2024-05-25

**Rating:** 3
**Confidence:** 2

**Summary:**

This paper investigates the structural memorization capabilities of large-scale text-to-image diffusion models and proposes a novel Membership Inference Attack (MIA) method based on structural similarity. Unlike traditional pixel-level approaches, the paper leverages structural similarity (SSIM) to distinguish between training set members and nonmembers. The experiments demonstrate the effectiveness and robustness of the proposed method across various conditions, including different image resolutions and distortions.

**Strengths:**

1.	The paper is well-structured, with a logical flow from the problem statement to the proposed solution and experimental validation, making it easy for readers to follow.
2.	The introduction of a structure-level membership inference attack, leveraging structural similarity (SSIM) rather than pixel-level noise comparison, represents an advancement in the field.

**Limitations:**

1. Clarification on Data Sampling in Section 3.2: The authors use 500 members and 500 nonmembers in Section 3.2. It should be clarified whether these two sets are sampled from the same distribution. Specifically, the authors should discuss whether the rate of decrease for these two groups would be more similar if they were sampled from the same distribution compared to being sampled from different distributions.

2. Lack of Discussion on Threshold $\tau$: The paper lacks a discussion on how the threshold $\tau$ is determined. The authors should include experiments that show how the threshold $\tau$ is set under different model architectures or datasets from different distributions, and whether there is a significant variation in the threshold value.

3. Limited Data Diversity: The experiments are primarily conducted on the LAION-400M and LAION2B-en datasets, which may not fully capture the method's performance across more diverse and specialized datasets such as medical or remote sensing images.

4. The font size in Figures 4 and 5 should be increased for better readability.

**Suitability:**

3

---

### Official Review · Reviewer_stxQ · 2024-05-30

**Rating:** 3
**Confidence:** 4

**Summary:**

Membership Inference Attack (MIA) is a hot topic for generative models, focusing on detecting whether the specific data sample is used in the model training. Compared with the traditional pixel-level MIA, this paper considers structure-level memorization: the structures of members are better preserved compared to those of nonmembers.

**Strengths:**

1. This paper matches the scope of ACM, and it will have a large group of audience since the security and privacy issues for generative models (especially for the multimedia domain) are quite important.
2. The authors well present the method, which is very easy to understand.

**Limitations:**

1. The idea of calculating structural similarity is quite straightforward.
2. The experiments are not adequate. The authors are expected to use more generative models (small size to large size), not limited to two models.
3.  Table 1 and Table 2 show that the proposed method cannot always achieve the best performance.

**Suitability:**

2

---

### Meta-Review · Area_Chair_tP7V · 2024-07-04

**Recommendation:** Accept (Poster)
**Confidence:** 3

**Metareview:**

This paper focuses on the membership inference attack task for diffusion-based generative models. Instead of using pixel-level information, the authors explore the role of structural information change during the diffusion procedure and propose using SSIM for this task. The method is clear and somewhat straightforward, but it was tested on four models (two of which are added in the supplementary materials), and the overall effectiveness is demonstrated.

After rebuttal, two reviewers provide their final ratings. One of them raised the rating from 3 to 4 and the other chose to keep the rating of 3 due to the unsolved issues. By reading this submission, the supplementary materials, the five reviews, and the rebuttal text, I think the overall contribution of this paper could benefit the development of research in this task and the remaining issues are not very hard to fix.